# Candidate Genes for Eyelid Myoclonia with Absences, Review of the Literature

**DOI:** 10.3390/ijms22115609

**Published:** 2021-05-25

**Authors:** Sonia Mayo, Irene Gómez-Manjón, Fco. Javier Fernández-Martínez, Ana Camacho, Francisco Martínez, Julián Benito-León

**Affiliations:** 1Genetics and Inheritance Research Group, Instituto de Investigación Sanitaria Hospital 12 de Octubre (imas12), 28041 Madrid, Spain; irenegomezmanjon@hotmail.com (I.G.-M.); ffernandezm@salud.madrid.org (F.J.F.-M.); 2Department of Genetics, Hospital Universitario 12 de Octubre, 28041 Madrid, Spain; 3Department of Neurology, Division of Pediatric Neurology, Hospital Universitario 12 de Octubre, Universidad Complutense de Madrid, 28041 Madrid, Spain; acamacho@salud.madrid.org; 4Traslational Research in Genetics, Instituto de Investigación Sanitaria La Fe (IIS La Fe), 46026 Valencia, Spain; martinez_fracas@gva.es; 5Genetics Unit, Hospital Universitario y Politecnico La Fe, 46026 Valencia, Spain; 6Department of Neurology, Hospital Universitario 12 de Octubre, 28041 Madrid, Spain; jbenitol67@gmail.com; 7Centro de Investigación Biomédica en Red sobre Enfermedades Neurodegenerativas (CIBERNED), 28031 Madrid, Spain; 8Department of Medicine, Universidad Complutense de Madrid, 28040 Madrid, Spain

**Keywords:** Jeavons syndrome, eyelid myoclonia with absences, candidate genes, *SYNGAP1*, *KIA02022*, *NEXMIF*, *RORB*, *CHD2*

## Abstract

Eyelid myoclonia with absences (EMA), also known as Jeavons syndrome (JS) is a childhood onset epileptic syndrome with manifestations involving a clinical triad of absence seizures with eyelid myoclonia (EM), photosensitivity (PS), and seizures or electroencephalogram (EEG) paroxysms induced by eye closure. Although a genetic contribution to this syndrome is likely and some genetic alterations have been defined in several cases, the genes responsible for have not been identified. In this review, patients diagnosed with EMA (or EMA-like phenotype) with a genetic diagnosis are summarized. Based on this, four genes could be associated to this syndrome (*SYNGAP1*, *KIA02022*/*NEXMIF*, *RORB*, and *CHD2*). Moreover, although there is not enough evidence yet to consider them as candidate for EMA, three more genes present also different alterations in some patients with clinical diagnosis of the disease (*SLC2A1*, *NAA10*, and *KCNB1*). Therefore, a possible relationship of these genes with the disease is discussed in this review.

## 1. Introduction

In 1977, Jeavons described something that is now known as Jeavons syndrome (JS): “Eyelid myoclonia and absences show a marked jerking of the eyelids immediately after eye closure and there is associated brief bilateral spike-and-wave activity. The eyelid movement is like rapid blinking and the eyes deviate upwards, in contrast to the very slight flicker of eyelids which may be seen in a typical absence in which the eyes look straight ahead. Brief absences may occur spontaneously and are accompanied by 3 c/s spike-and-wave discharges. The spike-and-wave discharges seen immediately after eye closure do not occur in the dark. Their presence in a routine EEG is a very reliable warning that abnormality will be evoked by photic stimulation” [1].

JS, also known as eyelid myoclonia with absences (EMA), is a childhood onset epileptic syndrome with manifestations involving a clinical triad of absence seizures with eyelid myoclonia (EM), photosensitivity (PS), and seizures or electroencephalogram (EEG) paroxysms induced by eye closure [2].

EMA is considered as a separate entity among genetic generalized epilepsies (GGE) associated with EM and brief absences related to generalized paroxysmal activity on EEG triggered by eye closure or intermittent photic stimulation (IPS) [3,4]. However, epilepsy with eyelid myoclonias has only been recently recognized as a distinct epilepsy syndrome by the International League Against Epilepsy (ILAE) [5].

A family history of seizures or epilepsy is common in those cases (seen in 40–80%) [4,5]. Additionally, reports of affected identical twins suggest its genetic etiology [4,6,7,8]. Despite this, there is actually no known gene accepted as pathogenic for this disease [5,9]. Moreover, different case reports have proposed several candidate genes [2,10,11,12,13].

EMA onset is typically in childhood, with a peak at 6–8 years. However, the time of seizure onset may be difficult to be exactly established, as eyelid jerks are frequently misinterpreted as tics or mannerisms, and absences may be overlooked [3]. The presence of massive myoclonus, intellectual disability (ID), or slowing of the EEG background are not typical features of the syndrome and may also cause delay in making the correct diagnosis [14]. More frequent in females, some patients show resistance to antiepileptic therapy [3,4].

In clinical practice, however, syndromes may overlap and cases may present with unusual manifestations, posing a diagnostic challenge [14]. The phenotypic and genetic heterogeneity may lead to underestimation of the clinical presentation, making the diagnosis more difficult [15]. In this review, based on different reported cases, we present four candidate genes for EMA and three other genes that might also be related to the disease. Moreover, we discuss their possible relation with the disease in order to improve the knowledge of this syndrome.

## 2. Results and Discussion

### 2.1. SYNGAP1

*SYNGAP1* (MIM *603384) is located on 6p21.32 [16]. This gene encodes a brain-specific synaptic Ras GTPase activating protein that is a member of the N-methyl-D-aspartate receptor complex [17]. Primarily expressed in excitatory neurons, it regulates dendritic spines structure, function, and plasticity, with major consequences for neuronal homeostasis and development, crucial for learning and memory [18]. Heterozygous loss of function variants in *SYNGAP1* are associated with developmental delay (DD), ID, epilepsy, and autism spectrum disorder (ASD) (MIM # 612621; ORPHA 544254) [19,20].

In 2011, Klitten et al. described a patient with epilepsy with myoclonic absences and a balanced translocation disrupting *SYNGAP1* [12]. This patient presented DD, ID, and ASD, but also eyelid winking and absences associated with eye deviation, being resistant to treatment (Table 1).

Mignot et al. (2016) presented a series of 17 unrelated patients with ID and epilepsy, mainly pharmacoresistant (>55%), carrying 13 different loss-of-function *SYNGAP1* mutations [21]. Three of them presented EM and suffered from seizures triggered by PS or EEG alteration after eye closure. Even more, one of them carry the missense alteration c.1685C>T (p.Pro562Leu), which has also been described in another patient diagnosed with EMA with myoclonic-atonic epilepsy (MAE) [22,23] (Table 1). However, this mutation was also recently reported by Lo Barco et al. (2021) in a patient without EM [24].

Vlaskamp et al. (2019) explored the relationship between EMA and MAE [23]. From a cohort of 57 cases with *SYNGAP1* mutations or microdeletions, the most common epilepsy phenotype was an overlapping syndrome combining the features of these two epilepsy syndromes (20/57, 35%), followed by the diagnosis of EMA in 13 patients (23%). DD/ID (32/33) and ASD (24/33) were also prevalent in those cases (Table 1). According to the results of Vlaskamp et al., absences with EM and PS were found in more than 50% of the cases with epilepsy and *SYNGAP1* alteration. Myoclonic (33%) and atonic (8%) seizures were also recurrent in their patients with EM. Moreover, in those cases, EM has an earlier onset and the cognitive outcome is worse than the classic syndrome of EMA. Therefore, they concluded that the more severe cases of EMA might be explained by *SYNGAP1* mutations, especially in those individuals with earlier onset of EM or myoclonic or atonic seizures [23].

In this sense, Kuchenbuch et al. (2020) presented three cases with different *de novo* mutations in *SYNGAP1* and epilepsy [25]. Although not all the clinical data are available, two of these cases presented EM and absences induced by PS or eye closure (Table 1). These two cases presented also myoclonic jerks and the EM onset was before 3 years of age (8 months and 2.5 years, respectively) [25]. In addition, the *SYNGAP1* variants of these two cases were also reported by Lo Barco et al. (2021) in two other cases with EMA (p.Glu656*) and EM (p.Arg687*) in a cohort of 15 patients with cognitive disability and pathogenic *SYNGAP1* variants, of which 14 were epileptic [24]. According to the clinical and EEG data, five of these patients presented EMA, with an onset age of three years or below, presence of myoclonic (60%) and atonic (40%) seizures in three and two cases respectively, and with uncontrolled seizures despite of the treatment in four cases. Moreover, two more cases of this series also presented EM and absences (Table 1) [24].

Finally, other publications gather more patients with alterations in *SYNGAP1* and a phenotype resembling EMA. Okazaki et al. (2017) published a case with a EMA-like phenotype that was carrier of a variant in this gene (p.Val1195Alafs*27) [26]. This alteration was previously identified in a male patient with moderate ID, no speech, psychomotor delay, and behavioral disorders, but without epilepsy [27]. However, this was also later described by Lo Barco et al. in a patient with EMA [24] (Table 1). Von Stülpnagel et al. (2019) published four cases with *SYNGAP1* pathogenic variants and EM typically initialed by eye closure [28]. It should be noted that two of them, which were also reported by Vlaskamp et al., were carrier siblings of a variant probably inherited from one of their parents as a result of gonadal mosaicism. Moreover, other variant (p.Leu323Pro) has also been described by Vlaskamp et al. in a patient with a very similar phenotype except for the photosensitivity (Table 1). Furthermore, four more cases of the series from Vlaskamp et al. were previously reported in different publications [22,29,30,31]. Therefore, a total of 49 patients with a phenotype of EMA or EMA-like, carriers of 44 different pathogenic variants in *SYNGAP1*, have been reported (Table 1).

### 2.2. KIA2022/NEXMIF

*NEXMIF* (MIM *300524), also known as *KIAA2022*, is located on Xq13.3 [32]. *NEXMIF* encodes for the X-linked Intellectual Disability Protein Related to Neurite Extension (XPN) [33]. Highly expressed in the early brain development, it participates in neurite outgrowth and regulates neuronal migration and cellular adhesion, critical for developing neuronal circuits [33,34,35]. Loss of function of *NEXMIF* causes mental retardation X-linked 98 (MRX98; MIM # 300912) [36]. Like most of X-linked disorders, males tend to be more severely affected than females, whereas carrier females present a wide phenotypic variability and may be unaffected as a result of random X-chromosome inactivation (XCI). MRX98 is a neurodevelopmental disorder characterized in males by delayed motor milestones, lack of language development, moderate to profound ID, behavioral abnormalities such as ASD, hypotonia, postnatal growth restrictions, dysmorphic facial features, and often early-onset seizures [36,37]. Compared with its hemizygous male counterpart, the heterozygous female disease has less severe ID, but is more often associated with a severe and intractable myoclonic epilepsy [38,39].

In 2017, Borlot et al. published the case of a women with EMA syndrome carrier of a *de novo NEXMIF* deletion of 77Kb, detected by genome-wide oligonucleotide array, within a cohort of 143 adults with unexplained childhood-onset epilepsy and ID [40]. One year later, Myers et al. reported two other sisters diagnosed with MAE with a point mutation at *NEXMIF* (p.Arg322*) in their search of parental gonadal mosaicism in apparently *de novo* epileptic encephalopathies [41]. These two cases and their clinical features have recently been reviewed by Stamberger et al. (2021) [42]. Analyzing the phenotype of 87 patients with *NEXMIF*-related encephalopathy, 10 females were diagnosed with EMA, 2 of them with an earlier onset (one year or younger), and 5 more cases (2 males and 3 females) presented a combination of EMA and MAE syndromes, including the case reported by Myers et al. (2018)) (Table 2) [42]. According to Stamberger et al., there was no correlation between phenotype and XCI status in their series, based on 1) the comparison of females with skewed and random XCI and 2) the families with sisters each presenting skewed and random XCI (families F4 and F7). However, it is interesting that in those families, both cases with a skewed XCI were diagnosed with MAE-EMA syndrome without photosensitivity, and their sister with a random inactivation presented a less severe phenotype. Moreover, XCI testing was performed in blood cells, so that the inactivation rate in neuronal cells is in fact unknown. Additionally, it is remarkable that from the two males, one presented the alteration in a 30% somatic mosaicism, which may lead to clinical repercussions equivalent to XCI in females (Table 2). On the other hand, the majority of patients with *NEXMIF*-related encephalopathy had drug-resistant epilepsy. Although specific information for each patient was not available, only 16% of the patients from the total cohort were seizure-free. It is outstanding that only 7% of the females, compared to 47% of males, were seizure-free (*p* = 0.001 Fisher’s exact) [42].

Finally, two more female patients with alterations in the *NEXMIF* gene and a phenotype resembling EMA have been reported. Wu et al. (2020) described a woman with refractory epilepsy and EEG features similar to those described in EMA who was a carrier of a nonsense variant in *NEXMIF* (p.Leu355*) (Table 2) [43]. Samanta and Willis (2020) identified a frameshift mutation (p.Asp573Serfs*11) in a girl with intractable seizures diagnosed with EMA [2]. She presented a XCI classified as random (ratio 74:26) (Table 2). Based on the results of Viravan et al. (2011) on occipital lobe relation to eye movements in JS, Samanta and Wills proposed that functional brain mosaicism, as a result of random XCI, causes a cellular interference effect responsible for the variable symptoms, with a predominant involvement of a circuit encompassing the occipital cortex and the cortical/subcortical systems physiologically involved in the motor control of eye closure and eye movements [2,44].

Summarizing, a total of 15 female patients and 2 males (one of which was a mosaic for the alteration) have been reported with pathogenic variants in *NEXMIF* and clinical features of EMA (Table 2).

### 2.3. RORB

*RORB* (MIM * 601972) is located on 9p21.13 [45]. This gene encodes for a nuclear receptor, retinoid-related orphan receptor β (RORβ), involved in neuronal migration and differentiation [46]. Recent evidences have point out that mutations in this gene may contribute to susceptibility to epilepsy (MIM # 618357) [47].

In 2012, Bartnik et al., within a cohort of 102 patients, described a case with epilepsy and EM with generalized tonic-clonic seizures (GTCS), carrier of a 2.57 Mb deletion of 6 genes including *RORB* [48]. A few years later, Rudolf et al. (2016) described a family with four affected family members of EMA with rare GTCS carriers of a nonsense variant in *RORB* (p.Arg66*) [49]. Other sporadic cases were also reported by these authors with different alteration on *RORB*, including two more cases with absences, EM and GTCS (Table 3). Sadleir et al. (2020) identified four novel RORB variants in 11 affected patients from four families with different epileptic syndromes [50]. Of this series, one case was diagnosed with EMA and occipital lobe epilepsy, presenting also GTCS. Moreover, another patient from a different family also presented absences with EM and GTCS, but was diagnosed with juvenile absence epilepsy and idiopathic photosensitive occipital lobe epilepsy (Table 3). Although the predominant epileptic phenotype of this cohort was represented by the overlap of photosensitive generalized and occipital epilepsy, the authors underlined the important role of occipital cortex in starting epileptic discharge in idiopathic generalized epilepsies such as EMA [50]. Finally, Morea et al. (2021) described another case with a *RORB* variant diagnosed with EMA [11]

Even though only six patients with *RORB* alterations, from three different families, have been clearly diagnosed with EMA, interestingly, five of them also presented GTCS. Moreover, four more patients presented EM and absences with GTCS.

### 2.4. CHD2

*CHD2* (MIM *602119) is located on 15q26.1 [51]. It encodes a member of the chromodomain helicase DNA-binding (CHD) family of proteins, of which the canonical function is the gene expression regulation by epigenetic changes in chromatin [52]. Loss of function of *CHD2* is identified as a cause of developmental epileptic encephalopathy (DEE) [52], being associated with childhood-onset epileptic encephalopathy (EEOC; MIM #615369) and MAE (ORPHA 1942) [53,54]. Usually, it is also characterized by cognitive regression, ID, ASD-like phenotype, and resistance to antiepileptic drugs (AED) treatment [52].

Two publications of 2015 underline the association of *CHD2* variants with photosensitivity in epilepsy, with seven patients with EMA between both articles [55,56]. Thomas et al. presented four cases with EMA, out of 10 patients with *de novo CHD2* alterations [56]. The four cases also have GTCS; in addition, other common features associated to *CHD2* deficiency were present (ID (4/4), ASD (3/4) and regression (3/4)) (Table 4). On the other hand, Galizia et al. presented the results of a *CHD2* screening in a series of more than 500 patients with photosensitive epilepsy [55]. From 36 patients with EMA, all with photoparoxysmal response, three cases presented unique variants in *CHD2* (Table 4). Based on the highest frequency of alterations among EMA patients compared to the rest of the series (8/544), the authors considered *CHD2* as an important contributor to EMA [55].

Although the number of reported cases with EMA and pathogenic variants in *CHD2* is low, it should also be considered in the screening for the genetic causes of this pathology.

### 2.5. Other Genes of Interest

Three publications present different patients with clinical diagnosis of EMA and with a pathogenic variant in three candidate genes for the disease: *SLC2A1*, *KCNB1*, and *NAA10*. The possible implication of these genes in EMA is discussed below.

*SLC2A1* (MIM *138140) is located in 1p34.2 and encodes for the major glucose transporter in the brain, GLUT1 [58]. It is responsible for the well-known GLUT1 deficiency syndrome and encephalopathy characterized by a childhood-onset epilepsy refractory to treatment, but with a wide phenotypic variability (MIM #606777; ORPHA 71277) [59,60]. In this sense, Madann et al. reported a pathogenic variant in *SLC2A1* in a family with Glut1-deficiency syndrome and JS [13]. The index case was a 9-year-old boy with intractable seizures since 4 months of age, and frequent absences with EM since 3 years of age. EEG showed eye closure sensitivity (eye closure triggered eyelid myoclonia with absences) and photosensitivity suggestive of EMA. He also presented multifocal seizures and paroxysms of intermittent involuntary gaze; sleep EEG showed multifocal interictal discharges and MRI was normal. Moreover, he had DD, mild ID, gait ataxia, scanning speech, and microcephaly. His father had a history of infantile-onset generalized epilepsy with generalized tonic-clonic seizures and ID, and his paternal uncle also had childhood-onset epilepsy. Metabolic results were suggestive of Glut1-deficiency syndrome; therefore, *SLC2A1* was sequenced. A pathogenic variant was detected in both the index case and his father (c.376C>T; p.Arg126Cys), in a hotspot located at a transmembrane domain of the GLUT1, that had been previously reported in other cases with the metabolic syndrome and typical absence seizures or myoclonic absences as the most prevalent seizure type but without EM or EMA features [61,62]. After different unsuccessful treatments with AEDs (valproate, phenobarbital, benzodiazepines, phenytoin, and topiramate), once the molecular diagnosis was known, a ketogenic diet allowed complete seizure remission. However, since it was a targeted study, other genetic causes in the index case could contribute to or be responsible for the EMA phenotype. Furthermore, a screening study of *SLC2A1* performed at 25 GGE-EM patients, including 8 cases of EMA, did not identify any variant that could confirm the role of *SLC2A1* in EMA or other GEE with EM [63]. Based on these cases, although EMA could be included within the wide phenotypic spectrum for non-classic GLUT1 deficiency syndrome, more evidence is required.

*KCNB1* (MIM *600397) is located in 20q13.13 [64]. It encodes for a brain potassium channel (Kv2.1) and its alteration causes a developmental epileptic encephalopathy (DEE26) (MIM # 616056) [65]. In 2017, from a cohort of six patients with *de novo* mutations in *KCNB1*, Marini et al. described two patients with a phenotype resembling EMA. The first case (patient 3) was carrier of a missense variant and was diagnosed with JS [66]. This patient was a 22-year-old female who had had epilepsy since 6 months of age, with bilateral myoclonic jerks. From 7 years of age, she developed absences with EM, frequently on eye closure or autoinduced, with persistent generalized PS. EEG showed generalized spike- and polyspike-wave discharges with a prominent generalized photoparoxysmal response, several episodes of myoclonia and absences with EM were recorded. She also presented myoclonic and tonic-clonic seizures. Trialed with several AEDs (Carbamazepine, valproic acid, levetiracetam, lamotrigine, ethosuximide, clonazepam, and topiramate), she finally became seizure-free with a combination of three of them. She had a delayed early development, evolving into mild cognitive impairment with motor and verbal dyspraxia, poor coordination, and moderate ID [66]. Her *KCNB1* variant (c. 916C>T; p.Arg306Cys) was located in the voltage-sensor domain of the protein, which was previously reported in a patient with DD and infantile-onset seizure refractory to therapy but without EMA [67]. A new case with this mutation was also recently reported by Minardi et al. (2020) in a series of 71 patients with DEE [68]. However, few specific clinical data for this case were provided, and EM or EMA-like features were not included among them. On the other hand, the series reported by Marini et al. included a second case with generalized epilepsy with myoclonic seizures and EM with PS; however, unfortunately, this patient was not clearly identified in the article [66]. Therefore, although the data reported by Marini et al. were promising, stronger evidence and casuistry are required to consider this gene as a candidate for EMA.

Finally, *NAA10* (MIM *300013) is located in Xq28 [69]. It encodes for an N-acetyltransferase and it is responsible for the Ogden syndrome in male carriers, a rare syndrome characterized by postnatal growth failure, developmental delay, hypotonia, and variable dysmorphic features (MIM # 300855) [70]. Although epilepsy is not associated to this syndrome, Valentine et al. (2018) describe a case with JS and a *de novo* variant in this gene [10]. The female patient, at the age of 3 years, presented initial seizures described as eye rolling and blank stares without generalized or focal body twitching. At first, she was diagnosed with absence epilepsy. Her seizures were frequently triggered by light stimulation. EEG showed a photoparoxysmal response, characterized by generalized spike-and-slow wave discharges, and numerous eyelid myoclonias with or without absences were recorded. Moreover, her seizures were intractable despite of the AED treatment (clonazepam, levetiracetam, lamotrigine, valproic acid, topiramate, rufinamide, clobazam, intravenous immunoglobulin, modified Atkins diet, and vagal nerve stimulator). Therefore, her epilepsy was consistent with EMA. She also presented DD, normal growth, self-injurious behavior and stereotypies, mild generalized hypotonia, and mild dysmorphisms (clinodactyly, mild ptosis, down slanting palpebral fissures, and tented upper lip) [10]. This patient’s *NAA10* variant (c.346C>T; p.Arg116Trp) was previously reported in a female patient with random XIC and without known seizure activity, but with other clinical features (normal growth, moderate ID, hypotonia, attention deficit hyperactivity disorder (ADHD), and developmental coordination disorder) [71]. This variant was also reported by Popp et al. (2015) in a male patient with a more severe phenotype (postnatal growth retardation, severe ID, truncal hypotonia and hypertonia of extremities, autistic features, and aggressive behavior) [72]. Moreover, EEG under photic stimulation of this reported male showed generalized epileptic form activity. However, the clinical differences might be due to the inactivation pattern of X chromosome in females, as commented for *NEXMIF* above.

More cases need to be collected to be able to consider these genes as candidates for EMA. However, in the next-generation-sequencing era, the screening for alteration in those genes in EMA-like patients is manageable and will allow to clarify their promising role in the disease.

### 2.6. General Overview of Genetic Interactions

As mentioned before, the seven genes are expressed in the brain and their function are of great relevance for neuronal development, migration, function, or genetic regulation; however, there is no clear relationship among them. Looking for possible interactions, *NEXMIF*, highly expressed in fetal and adult brain [73], might be related to *RORB*, involved in neuronal migration and differentiation [46], and to *SLC2A1*, essential to provide the requirements of glucose at the brain among other tissues [74] (Figure 1) [75]. However, further studies, including in vitro and animal model assays, would be required to confirm this hypothesis.

### 2.7. Animal Models

Gene editing techniques have facilitated the generation of mouse models of human diseases; however, very little is known specifically about EMA. *SYNGAP1* haploinsufficient young mice showed a reduced fluorothyl-induced seizure threshold and were prone to audiogenic seizures [76]. Furthermore, it is worth noting that in the former study, photostimulation evoked signals originating in the dentate gyrus were dramatically amplified as they spread through the hippocampus, instead of attenuated as it occurs in wild type animals. In addition, germline *Syngap1* mutations in mice induced a persistent form of stabilized cortical hyperexcitability that lasted into adulthood, with the seizure threshold remaining reduced [77]. Interestingly, restoration of the gene in adult mice was able to improve behavioral and electrophysiological measures of memory and seizures [78]. Finally, the phenotype of the epileptogenesis in a *Syngap1*^+/−^ mouse model have been recently described [79]. On the other hand, loss of *NEXMIF* gene expression in neurons of Knock-out (KO) mice results in a significant decrease in synapse density and synaptic protein expression [80]. These animals presented severe seizures, although further studies are required to characterize the epileptic phenotype in Nexmif KO mouse models.

## 3. Methods

Systematic literature research of PubMed was performed to identify eligible articles until 31 March 2021 (see Appendix A for complete search terms). The search identified 66 potential articles.

Reviews, clinical trials, and articles in a different language than English were excluded. We screened the titles and abstracts to check if they were within the scope of this review. In some cases, when abstracts were not available or more information was required to decide, a quick review of the whole article was carried out. At this stage, a total of 21 original articles, mainly case reports, containing data dealing with candidate genes for EMA, were obtained (Figure 2). For each of the seven selected genes, a literature research was also performed to look for more cases with an EMA-like phenotype. This selection was based on different number of cases for each gene: 49 for *SYNGAP1* (Table 1); 17 for NEXMIF (Table 2); 10 for RORB (Table 3); 7 for CHD2 (Table 4); 2 cases for KCNB1; and 1 case each for SLC2A1 and NAA10. It is also remarkable that some of these cases were reported in different publications (Table 1, Table 2 and Table 4).

## 4. Conclusions

Loss of function of *SYNGAP1*, in addition to its association with DD, ID, and ASD, might be considered in epileptic patients with EMA, especially in those cases with earlier onset of EM, pharmacoresistance, or myoclonic or atonic seizures.

The phenotype spectrum of *NEXMIF* in females (or mosaic males) may also include EMA-like, probably associated with pharmacoresistance. Since this gene is located in the X chromosome, XCI in the brain can causes specific cellular mosaicism that might be responsible for the EMA phenotype in those cases.

In patients with alterations in *SYNGAP1* or *NEXMIF*, clinical features of EMA may overlap with MAE syndrome, presenting manifestations of both pathologies. It is also remarkable that despite the relative low number of cases with pathogenic alteration in those genes, two families presented a probably gonadal mosaicism: one family presented with a frame shift mutation in *SYNGAP1* (p.Leu150Valfs*6) and the other with a nonsense variant in *NEXMIF* (p.Arg322*) (Table 1 and Table 2). The recurrence of gonadal mosaicism is very variable depending on the disease but has to be taken into account for correct genetic counseling [81].

Regarding *RORB* and *CHD2*, although the number of cases with EMA is significantly lower, they should be taken into account, especially in those cases with GTCS.

In relation to other genes, a few cases of EMA have been reported with variants in *SLC2A1*, *KCNB1*, or *NAA10*. There is not enough information to establish a clear relationship, but as more and more exome and genome studies in EMA patients are performed, it is expected that their role in the molecular diagnosis of this pathology will be clarified.

It is remarkable that two of the seven genes are located in the X chromosome, *NEXMIF* and *NAA10*. Although males show an apparently more severe phenotype, females more often present severe and intractable epilepsy. As mentioned before, random XCI in the brain might lead to cellular interference responsible for the epilepsy and could also explain the higher prevalence of EMA in females.

Finally, an animal model is a great tool to study the pathobiology of complex human disease that affect organs such as the brain. Although mouse models have shown some results regarding *SYNGAP1* and *NEXMIF* haploinsufficiency, no specific data have been collected for EMA. Therefore, to establish the possible interaction between these seven genes, or their direct implication into the pathology, further functional studies are required.

## Figures and Tables

**Figure 1 ijms-22-05609-f001:**
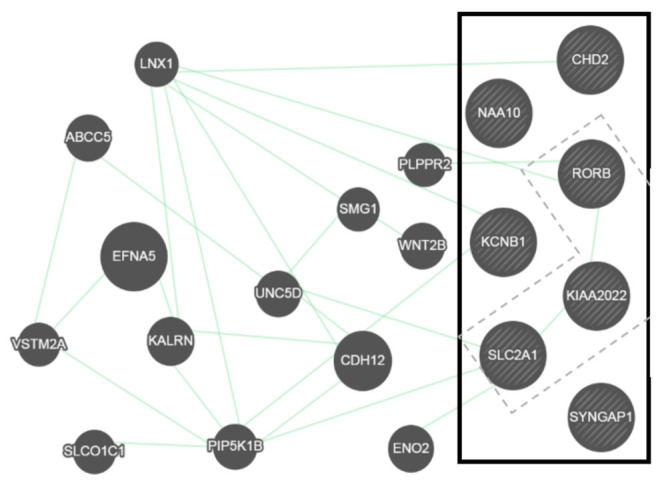
Prediction of the genetic interactions performed by Genemania (http://genemania.org/ (accessed on 15 April 2021) [75].

**Figure 2 ijms-22-05609-f002:**
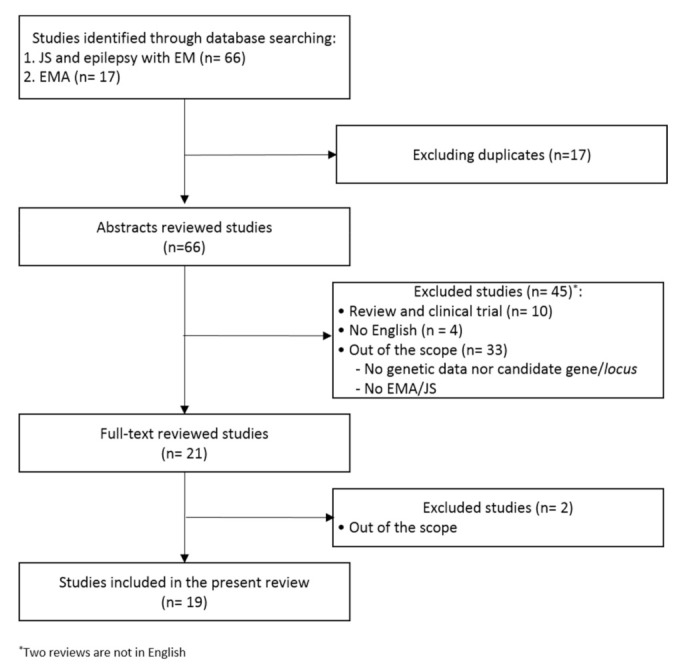
Flow diagram summarizing the systematic search, screening, and studies selection for this review.

**Table 1 ijms-22-05609-t001:** Summary of the cases reported with pathogenic (or probably pathogenic) alteration in *SYNGAP1* (NM_006772) and an EMA/EMA-like phenotype.

A																												
	Reference		Klitten 2011 ^1^			Mingot 2016 ^a^				Mingot 2016			Mingot 2016			Okazaki 2017 ^b^					Vlaskamp 2019 ^1^			Vlaskamp 2019 ^1^		Vlaskamp 2019 ^1,2^	Vlaskamp 2019 ^1,2^	
	Patient					11				12			14								1			4		5	7	
Clinical features	gender		Male			Male				Female			Female			Male					Male			Female		Male	Male	
	Age		25 yr			3 yr				22 yr			8 yr			4 yr					3 yr 10 months			8 yr 3 months		11 yr 2 months	17	
	DD/ID		+ (Severe)			+ (Severe)				+ (Severe)			+ (Mild)			+					- (FSIQ: 80, low average)			+ (Mild)		+ (Moderate)	+ (Severe)	
	Behavioral features		ASD traits, anxious behavior			Repetitive behaviours, stereotypies				Stereotypies			ASD, stereotypies			NR					-			ASD traits, mild tantrums, aggression, high pain thresholds, sleeping problems		-	ASD, regression severe tantrums, self-injury, aggression, high pain threshold, sleeping problems	
	Other parametres		Absence of language			Absence of language, truncal hypotonia, swallowing difficulties				Absence of language, mild gait ataxia, flexion deformity of left hip, hyperlordotic lumbar spine, microcephaly			Motor slowness and moderate akinesia, ataxic gait, truncal hypotonia, dystonic postures of hands and feet, plastic hypertonia			Hypotonia, hypersalivation					-			Hypotonia		-	Hypotonia, unsteady gait, reflux, obstipation, eating difficulties, benign bone tumor	
Epilepsy	Age of onset		13 months			2 yr				1 yr			5 yr			1 yr and 5 months					16 months			2.5 yr		11 months	2 yr	
	Abscence seizures		+ (MA, AA)			NR				+ (AA)			+ (MA)			NR					+ (MA)			NR		NR	NR	
	Eyelid myoclonia		+ (eyelid winking)			+				+			+			NR					+			+		+	+	
	Photosensitivity		NR			+				NR			+			+					+			+		+	+	
	Other seizures		DA with MJ			FS				FS, MJ			NR			Upward eye deviation, motion arrest, loss of consciousness, and eyelid twitching. Triggered by crying and photosensivility					MS. Triggered by PS, sounds, sleep deprivation and fatigue			MS		FS, MAt, bilateral TCS. Triggered by PS, sleep deprivation and fatigue	Bilateral TCS, MS, FIAS, DA. Triggered by PS, eye closure and eating	
EEG	others		Interictal: generalized synchronous 3–7 Hz (P)SW			Abnormal BG, generalized slowing, EM, and generalized seizure patterns. Triggered by photosensivility				Bursts of spikes and slow waves in the occipital region after eye closure. Triggered by FOS			Ictal: bursts of diffuse PSW with posterior predominance after eyes closer and photic stimulation. Triggered by FOS and PS			Ictal: diffuseslow or SW activity with occipitalto central predominance. Interictal: bilateral frontal spikes. Sleep: rhythmic, generalized 2–3-Hz delta activity, without visible seizures. Normal BG					Ictal: GSW (myoclonic). Interictal: GSW			BG slow. Interrictal: Frequent 2.5–4 Hz GSW, after eye closure in trains, MFD		Interictal: GPW	sleep: frequent seizures while falling asleep	
	Cranial MRI		NR			Normal				Normal			Normal			Normal					Normal			Normal		Normal	NR	
	AED Treatment		VPA, LTG, CLB			VPA				LEV, TPM			LEV, ETX			CBZ, VPA, LEV, ETX, LTG					VPA			VPA, LEV, LTG, ETX, CBD		VPA	VPA, CLZ, CBZ	
Geneticinformation	Genetic test		karyotype			gene panel NGS				WES			WES			NGS panel					NR			NR		NR	NR	
	Genomic change (Hg19)		NR			chr6:33406650; C>C/T				chr6:33408514; C>C/T			chr6:33409458_ 33409461delAGCG			chr6:33414346; G>G/A					chr6:33391277; C>C/T			chr6:33399974; CA>CA/C		chr6:333400498_ 33400501delAAAC	chr6:33400501; C>C/T	
	cDNA/aa change		(truncate gene) ish, 46,XY, t(6;22)(p21.32;q11.21)dn			c.1630C>T, p.Arg544*				c.1685C>T, p.Pro562Leu			c.2214_2217delAGCG, p.Glu739Glyfs*20			c.3583-6G>A, p.Val1195Alafs*27					c.91C>T, p.Arg31*			c.333delA, p.Lys114Serfs*20		c.424_ 427delAAAC, p.Lys142Glufs*31	c.427C>T, p.Arg143*	
	Inheritance		de novo			de novo				de novo			de novo			(parent not tested)					de novo			de novo		de novo	de novo	
	Others information		FISH (probe RP11.497A24)			VUS inherited from the mother: SCN9A: c.4282G>A and c.5624G>A; ARX: c.1462A>G				-			-			Karyotype and anlysis for AS with normal results. CSF glucose normal												
	Family History		- (epilepsy)			-				-			-			- (epilepsy or ID)					- (seizures, ID or ASD)			- (seizures, ID or ASD)		- (seizures, ID or ASD)	Sister, father, and many paternal relatives with learning difficulties. Maternal cousin ASD	
	**B**																											
	**Reference**		**Vlaskamp 2019 ^1,2^/Von Stülpnagel 2019**			**Vlaskamp 2019 ^1,2^/Von Stülpnagel 2019**	**Vlaskamp 2019 ^1,2^**				**Vlaskamp 2019 ^1,2^**				**Vlaskamp 2019 ^1,2^/The DDD Study 2017**		**Vlaskamp 2019 ^1,2^**				**Vlaskamp 2019 ^1^**			**Vlaskamp 2019 ^1^**		**Vlaskamp 2019 ^1,2^**	**Vlaskamp 2019 ^1,2^**	
	**Patient**		**8/7**			**9/8**	**11**				**12**				**13/241**		**15**				**17**			**18**		**20**	**21**	
Clinical features	gender		female			female	male				female				male		female				female			male		female	female	
	Age		8 yr			6 yr	6 yr				4 yr 8 months				7 yr 10 months		9.5 yr				5 yr 2 months			5 yr 10 months		15 yr 1 months	10 yr 5 months	
	DD/ID		+ (moderate-severe)			+ (moderate-severe)	+ (severe)				+ (severe)				+ (severe)		+ (severe)				+ (severe)			+ (severe)		+ (moderate)	+ (moderate)	
	Behavioral features		regression, tantrums, self-injury, tichotillomania, high pain threshold, eating disorder, sleeping problems			ASD, echolalia, high pain threshold, eating disorder, sleeping problems	aggression, high pain threshold				regression, ASD, tantrums, self-injury, aggression. high pain threshold, sleep problems				Tantrums, aggression, high pain threshold, sleep problems		ASD traits, stereotypes, odontoprisis, high pain threshold				odontoprisis, high pain threshold			regression, obsession, self-injury, aggression, high pain threshold, sleep problems, eating disorder		ASD, self-injury, aggression, sleep problems, high pain threshold, oral hypersensitivity	regression, ASD, severe tantrums, self-injury, aggression, sleep problems, high pain threshold, oral hypersensitivity	
	Other parametres		hypotonia, hyperlaxity, 2 café au lait spots, small capillary hemangioma, constipation, hearing loss after infection/ataxia, problems in fine motor skills			hypotonia, hyperlaxity, hearing loss after recurrent otitis/ataxia					hypotonia, ataxia, constipation, reflux, absence of language				pes planus, strabismus, constipation, hearing loss		nystagmus				Congenital hipdyslocation, absence of language			hypotonia, poor balance and coordination, absence of language		hypotonia, hypermobility, scoliosos, constipation	congenital nystagmus, hypotonia, a few cafe au lait macules, constipation	
Epilepsy	Age of onset		8/16 months			12–13 months	4 yr				23 months				2 yr		2 yr				3.5 yr			3 yr		12–14 months	2 yrs	
	Abscence seizures		+			+	NR				+ (AA)				NR		NR				NR			NR		+ (MA)		
	Eyelid myoclonia		+			+	+				+				+		+				+			+		+	+	
	Photosensitivity		+			+	NR				NR				NR		NR				+			NR		+	+	
	Other seizures		MS, MAt. Triggered by touch and thinking of eating/GS, MJ, atonic drops. Reflex seizures while chewing			MS, MAt. Triggered by thinking of eating/GS, MJ, atonic drops. Reflex seizures while chewing	MS, DA. Triggered by fever and infection				MS, AS				MS, AS, bilateral TCS. Triggered by eating (chocolate), fever and fatigue		Bi- and unilateral TCS				MJ, MS			Triggered by fatigue		atonic DA, nocturnal TS. Triggered by PS and eating	MJ, bilateral TCS	
EEG	others		Intal: 3 Hz GSW (EM-MAt). Interictal:G(P)SW			Intal: 2.5–3.5 Hz GSW (EM-MAt). Inerictal: 2.5–3.5 Hz GSW	BG poor. Ictal: Bilateral occipital sharps, followed by MFD (EM). Interictal: MFD				BG: slow. Ictal: GPSW (MS), 1.5–2 Hz GSW (AA). Interictal: GSW facilited by eye closure				Interictal: GSW		BG: slow. Ictal: FD (unilateral TCS). Interictal: 3–4 Hz GSW, MFD				BG: slow. Interictal: 1.5–3 Hz GSW, MFD			Interictal: 3 Hz GSW, bifrotal SW		BG: slow. Ictal: GSE (MA). Inerictal: MFD	BG: slow. Interictal: 3 Hz GSW, also following eye closure, FD	
MRI	Cranial		Normal			Normal	Normal				Normal				Normal		Normal				Normal			Normal		Normal (discrete hippocampal tissue loss, not progressive and without sclerosis)	Normal	
Treatment	AED		VPA, LEV, TPM, CLB, LTG, ETX, LCM, ZNS, CLZ, CBD, PHT			CLB, LCM, ZNS, CLZ, CBD	VPA, LEV, TPM				CLB, TPM, NZP, LTG, VPA, CLZ				VPA, CLB		VPA, CLZ, LEV, TPM				VPA, CLZ, LEV, CLB			VPA		VPA, LEV, LTG	VPA	
	Other		KD			KD, mAD											Vitamin B6											
Genetic information	Genomic change (Hg19)		chr6:33400509_33400521dup				chr6:33403058delC				chr6:33403318dupC				chr6:33403367; C>C/T		chr6:33405511_33405512insC				chr6:33406048; C>C/T			chr6:33406202delC		chr6:33406324;C>C/G	chr6:33408547G>GGCTGC	
	cDNA/aa change		c.435_447dup, p.Leu150Valfs*6				c.639delC, p.Ile214Trpfs*9				c.690dupC, p.Phe231Leufs*14				c.739C>T, p.Gln247*		c.822_823insC, p.Lys277Glnfs*7				c.1366C>T, p.Gln456*			c.1393delC, p.Leu465Phefs*9		c.1515C>G, p.Tyr505*	c.1718_1719insGCTGC, p.Glu578Alafs*74	
	Inheritance		de novo/mosaic parent				de novo				de novo				de novo		de novo				de novo			de novo		de novo	de novo	
	Family History		sisters				- (seizures, ID or ASD)				- (seizures, ID or ASD)				Maternal aunt and distant relative epilepsy, other distant relatives ASD		- (seizures, ID or ASD)				- (seizures, ID or ASD)			Maternal grandfather post-stroke epilepsy		Paternal uncle moderate ID	- (seizures, ID or ASD)	
	**C**																											
	**Reference**		**Vlaskamp 2019 ^1,2^**			**Vlaskamp 2019 ^1,2^**		**Vlaskamp 2019 ^1^**			**Vlaskamp 2019 ^1^/Carvill 2013**			**Vlaskamp 2019 ^1^**				**Vlaskamp 2019 ^1^**			**Vlaskamp 2019 ^1,2^**			**Vlaskamp 2019 ^1^**	**Vlaskamp 2019 ^1^**		**Vlaskamp 2019 ^1^**	
	**Patient**		**23**			**24**		**25**			**26/T2528**			**27**				**30**			**31**			**32**	**33**		**35**	
Clinical features	gender		female			female		male			male			male				female			male			female	male		male	
	Age		11 yr 11 months			7 yr		11 yr 7 months			30/26 yr			6 yr				3 yr 11 months			11 yr 2 months			33 yr	15 yr 3 months		4 yr 9 months	
	DD/ID		+ (severe)			+ (severe)		+ (moderate)			+ (moderate)			+ (severe)				+ (severe)			+ (severe)			+ (moderate-severe)	+ (severe)		+ (moderate-severe)	
	Behavioral features		regression, ASD, tantrums, self-injury, aggression. Sleep problems, high pain threshold, eating difficulties			regression, ASD, aggression, sleep problems, high pain threshold		ASD, self injury, aggression, sleep problems, high pain threshold			regression, OCD symptoms, tantrums, aggression			regression, ASD, high pain threshold				ASD, tantrums, self-injury, aggression, sleep problems, high pain threshold, oral hypersensibility			regression, ASD, tantrums, aggressive, sleep problems, high pain threshold, eating difficulties			regression, ASD, self-injury, aggression, poor concentration, high pain threshold	regression, ASD, aggression, sleep problems, high pain thresholds, eating difficulties		ASD, aggression, sleep problems, high pain threshold	
	Other parametres		hypotonia, constipation			congenital hisdysplasia, hypotonia, ataxic gait		pes planus, hypotonia, unsteady gait, constipation			mild two/three syndactyly, irregular tremor upper extremities, osteopenia			Unsteady gait				hypotyonia, unsteady gait, constipation, chronic idiopathic tromnocytopenic purpura, absence of language			microcephaly, short stature, borderline hypotonia, ataxia, Hemangioma nasal cavity			Hypotonia, coordination disoder/ataxia	right pes planus, left pes caves, hypotonia, bilateral pyramidal syndrome, unsteady gait, orthothics, hyperflexibility		mild hypotonia	
Epilepsy	Age of onset		2 yr			6 months		9.5 yr			18 months			2 yr				18 months			2 yr 3 months			8 months	18 months		2 yr 1 month	
	Abscence seizures		NR			NR		NR			+			NR				NR			NR			NR	NR		NR	
	Eyelid myoclonia		+			+		+			+			+				+			+			+	+		+	
	Photosensitivity		NR			NR		NR			+			NR				NR			NR			+	NR		NR	
	Other seizures		FS, bilateral TCS (with fiver)			bilateral TCS (with fiver), atonic DA. Triggered by fatigue and illness		Triggered by eating			FS, aura, FIAS, MJ, NCSE, bi- and unilateral TCS, MS. Triggered by PS			Triggered by hunger, self-induced with hyperventilation, fatigue and stress				TCS (with fiver)			Atonic DA. Triggered by sounds, fatigue, and drop in emperature			GTCS. Triggered by PS	-		Triggered by eating	
EEG	others		IBG: Slow. Ictal: 2–3 Hz GSW with frontal maximum, (EM-AS). Interictal: MFD			Interictal: 2.5 Hz GSW		NR			BG: slow. Interictal: occipital 2 Hz GSW, occipital FD/bi-occipital ED, DS, SSW			BG: iregular. Ictal: GSW (EM). Interictal: 2–3 Hz GSW, irregular GPSW, MFD				interictal: epileptiform discharge			Ictal: irregular GSW followed by slower discharges (EM), GPSW (EM). Interictal: G(P)SW, bifrotal SW, FD			NR	BG: slow Ictal: GSW (EM). Interictal: temporo-occipital SW, 10% generalized activity in 24 hours.		BG: right occipital slowing. Ictal: eyeblink without ictal correlate. Interictal: only in sleep: right occipital slowing, focal sharp waves	
	Cranial MRI		Normal			Normal		Normal			Normal			Patent cavus vergae				Atypical WM abnormalities			Normal			NR	Enlarged ventricles		Normal	
Treatment	AED		VPA, LEV			VPA, LEV		-			VPA, LTG, CLB			VPA, LEV, ETX, ZNS, CBD, LTG				-			VPA, CLB, TPM, LTG			VPA, CBZ, TPM	VPA, LEV, LTG		-	
	Other													KD							KD							
Genetic information	Genetic test		NR			NR		NR			NGS panel			NR				NR			NR			NR	NR		NR	
	Genomic change (Hg19)		chr6:33409006;G>G/A			chr6:33409095;C>C/T		chr6:33409095;C>C/T			chr6:33409140;C>C/T			chr6:33409419_3349422del				chr6:33411265_ 33411267delinsCA			chr6:33411735dupC			chr6:33412317;G>G/T	chr6:33414426;T>T/G		chr6:33393573;A>A/G	
	cDNA/aa change		c.1970G>A, p.Trp657*			c.2059C>T, p.Arg687*		c.2059C>T, p.Arg687*			c.2104C>T, p.Gln702*			c.2177_2180delGGAA, p.Arg726Thrfs*33				c.2936_ 2938delinsCA, p.Phe979Serfs*98			c.3406dupC, p.Gln1136Profs*17			c.3505G>T, p.Glu1169*	c.3657T>G, p.Tyr1219*		c.190-2A>G, (splice acceptor site)	
	Inheritance		unknow			de novo		de novo			de novo			de novo				de novo			de novo			de novo	de novo		de novo	
	Family History		Distant relative ASD			Paternal grandmother GTCS 16–20 y. Distant relative ASD		- (seizures, ID or ASD)			Distant relative epilepsy			- (seizures, ID or ASD)				Maternal aunt ID. Paternal first cousin post-traumatic epilepsy			Maternal uncle ID post-meningitis. Distant relative epilepsy			Maternal and paternal first cousins learning difficulties	Distant relative ASD		- (seizures, ID or ASD)	
		**D**																										
		**Reference**		**Vlaskamp 2019 ^1,2^**	**Vlaskamp 2019 ^1^/Parrini 2017**	**Vlaskamp 2019 ^1^**		**Vlaskamp 2019 ^1,2,c^**				**Vlaskamp 2019 ^1,2^**		**Vlaskamp 2019 ^1,2^**					**Vlaskamp 2019 ^1,2,a^/Berryer 2013 ^a^**		**Vlaskamp 2019 ^1,2^**		**Vlaskamp 2019 ^1,2^**			**Von Stülpnagel 2019 ^2,c^**		**Von Stülpnagel 2019**
		**Patient**		**36**	**39/1190N**	**41**		**45**				**46**		**50**					**51/3**		**52**		**53**			**1**		**2**
Clinical features		gender		female	female	female		male				male		male					female		female		female			male		male
		Age		8 yr 11 months	16.4/17 yr	6 yr 8 months		3 yr 2 months				10 yr		15 yr 1 month					9.1/4.2 yr		8 yr 3 months		7 yr months			5 yr		14 yr
		DD/ID		+ (mild)	+ (moderate-severe)	+ (moderate-severe)		+ (severe)				+ (moderate)		+ (severe)					+ (moderate/mild)		+ (severe)		+ (mild)			+ (moderate)		+ (moderate-severe)
		Behavioral features		regression, ASD traits, tantrums, self-injury, aggresive	regression, Aggressive, REM sleep behavioral disorder, ASD	regression, ASD, sleep problems, high pain threshold		regression, ASD traits				regression, ASD, tantrums, OCD, echolalia, high pain threshold, eating dificulties		regression, ASD, eating disorder					regression, ASD, OCD, tantrums, self-injury, sleep problems, high pain threshold, eating disorder		regression, ASD, tantrums, self-injury, sleep problems, high pain threshold		tantrums, aggressive, sleep problems			regression, ASD		ASD
		Other parametres		Few café au lait macules	pronated foot, coordinarition disorder, ataxic gait	hypotonia		hypospadie, 6th toe, hypotonia, nystagmus				Mild cerebral palsy, pes planus, mild musle weakness, clinodactyly toes, hypotonia, constipation, coeliac disease		obesity, ataxia with wide-based gait					Hypotonia, unsteady gait with poor balance, gross and fine motor dyspraxia		macrocephaly, knee hyperextension, pes planus, pronated feet, hypotonia, wide-based gait with poor balance, constipation		pes caves, hypotonia, balance issues			Height <3p. Tongue hypotonia and horizontal nystagmus. Postaxial hexadactylia and hypospadia		Abnormal gait; poor coordination; dysarthria. Abnormal facial shape (triangular), large anteverted, ears, wide mouth, thin lips, pointed chin
Epilepsy		Age of onset		<2 yr	6.7/5 yr	4.5 ys		2.5 yr				2.8 yr		2.5 yr					18 months		18 months		2.5 yr			2.5 yr		20 months
		Abscence seizures		NR	NR	+ (typical)		NR				NR		+ (typical)					NR		NR		NR					
		Eyelid myoclonia		+	+	+		+				+		+					+		+		+			+		+
		Photosensitivity		NR	+	+		+				-		+					-		NR		-			-		+
		Other seizures		Triggered by eye closure, hunger and fatigue	Triggered by PS	-		MAt, MS, atonic DA. Triggered by eating and stress				Triggered by visual patterns		bilateral TCS (with fiver). Triggered by PS and noise					bilateral TCS, myoclonic DA. Triggered by eating		Triggered by eating, eye closure and fatigue		Triggered by illness and fatigue			Atonic head dropping		FS, GTCS, episodes characterized by loss of consciousness, backward eyeball rolling, MS; generalized with head atonia and EM; status epilepticus.
EEG		others		Ictal: GD (EM-MAt). Interictal:2–3 Hz GSW, frequent GD, induced by eye closure	Ictal: G(P)SW (EM)Interictal: G(P)SW	Ictal: G(P)SW (EM)		NR				BG: slowing. Interictal: GPS, 3.5–4 GSP, FD		Interictal: GD, MFD, spikes, G(P)SW					BG slow		Interictal: GSW		BG: gegeralized slowing. Interictal: MFD			1–3 s lasting high amplitude 3/s SW complexes with bilateral initiation and occipital predominance but never lateralized. Triggered by heat, fatigue, stress, and orofacial stimuli		slowed BG activity (theta); spikes and polyspikes over the occipital regions; abnormalities are worsened by sleep. Triggered by PS, autoinduced, and eating.
MRI		Cranial		Normal	Normal	Mega cisterna magna fossa posterior		Normal				Normal		Normal					Small hyperintens subcortical WM lesions (bi-frontal, peri-ventricular), possibly post-anoxic leukopathy		Stable mild enlarged ventricles and pineal cyst		Normal			Slight frontal dilatation of the external spaces of cerebrospinal fluid and an age-appropriate myelination		Normal
Treatment		AED		VPA, CLB, LTG	LZP, RUF, VPA	LEV, ZNS, RUF, VPA		VPA, LTG				VPA		Hydrocortison, VGB, NZP					VPA		VPA, LEV, ETX, CLZ, LTG, CBD		VPA, ZNS, LEV, PER, CBD			VPA, LTG		VPA, LTG, LEV, CLB
		Other		NR	NR	NR		NR				NR		NR					NR		NR		KD			NR		NR
Genetic information		Genetic test		NR	NGS panel (95 genes)	NR		NR				NR		NR					NR		NR		NR			gene panel NGS		NR
		Genomic change (Hg19)		chr6:33400029;G>G/A	chr6:33400583; G>G/A	chr6:33408504_ 33408514delAGCGTGTTCCC		chr6:33405650;T>T/C				chr6:33405712; G>G/A		chr6:33406199;T>T/ TTCC					chr6:33408514;C>C/T		chr6:33408626;C>C/G		chr6:33408718;T>T/A			chr6:33405650;T>T/C		chr6:33400462;C>C/T
		cDNA/aa change		c.387G>A, p.Ser129Ser (splice donor site)	c.509G>A, p. Arg170Gln	c.1677-2_1685del, (splice acceptor site)		c.968T>C, p.Leu323Pro				c.1030G>A, p.Gly344Ser		c. 1390delinsTTCC, p.Leu465dup					c.1685C>T, p.Pro562Leu		c.1797C>G, p.Cys599Trp		c.1889T>A, p.Ile630Asn			c.968T>C, p.Leu323Pro		c.388C>T, p.Gln130Ter
		Inheritance		de novo	de novo	de novo		de novo				de novo		de novo					de novo		de novo		de novo			de novo		de novo
		Others information																	Karyotype, aCGH, MECP2 mutation and X-fragile analysis, without relevant results									
		Family History		- (seizures, ID or ASD)	- (seizures, ID or ASD)	- (seizures, ID or ASD)		Paternal uncle epilepsy and behavioural problems				- (seizures, ID or ASD)		Mother FS. Paternal uncle learning difficulties					Maternal first cousin ASD		- (seizures, ID or ASD)		- (seizures, ID or ASD)			- (epilepsy or DD)		NR
	**E**																											
	**Reference**		**Kuchenbuch 2020 ^d^**			**Kuchenbuch 2020 ^e^**			**Lo barco 2021 ^e^**		**Lo barco 2021**			**Lo barco 2021**						**Lo barco 2021 ^d^**		**Lo barco 2021 ^b^**				**Lo barco 2021**		**Lo barco 2021**
	**Patient**		**1**			**3**			**2**		**4**			**5**						**6**		**8**				**9**		**10**
Clinical features	gender		male			male			male		male			female						male		male				female		female
	Age		5.6 yr			3.5 yr			3 yr 4 months		11 yr 3 months			6 yr 3 months						5 yr 4 months		6 yr 6 months				14 yr		7 yr
	DD/ID		NR			NR			+ (severe/moderate)		+ (severe)			+ (severe/moderate)						+ (moderate)		+ (severe)				+ (severe/moderate)		+ (severe/moderate)
	Behavioral features		motor stereotypies, heteroaggressivity, ASD			NR			ASD, behavior disorder		ASD, behavior disorder			ASD, behavioral and sleep problems						ASD, behavior disorder		ASD, feeding problems				ASD, behavioral and sleep problems		sleep, behavior and eating disorders
	Other parametres		NR			NR			NR		head growth slowdown, absence of language			growth delay, absence of language						NR		nystagmus, absence of language				head growth slowdown, absence of language		head growth slowdown, absence of language
Epilepsy	Age of onset		3.5 yr			8 months			8 months		27 months			18 months						36 months		20 months				20 months		30 months
	Abscence seizures		+ (AA)			+ (MA)			+ (AA, AAM)		+ (AA)			+ (AA, AAM, AAOC, AAA)						+ (AA, AAOC, MA)		+ (AAM)				+ (AA, AAM)		+ (AAM)
	Eyelid myoclonia		+			+			NR		+			+ (EMA)						+ (EMA)		NR				NR		+ (EMA)
	Photosensitivity		+			NR			NR		NR			NR						NR		NR				NR		NR
	Other seizures		FS, MS, upper limb MJ, DA. Triggered by sleep and IPS			DA, upper limb MJ, reflex seizures self-induced by eye closure			MS, FS		DA			AS						NR		MS, TS				MS		NR
EEG	PPR		NR			NR			+		-			+						+		-				+		-
	others		NR			2-HZ GPSW			Sleep: Sporadic low-voltage multifocal spikes; sporadic bursts of generalized irregular polyspike or PSW in sleep. EM, AA, AAM, F. Self stimulation wirh eyes closure.		Wake: (P)SW on frontal regions; Sleep: higher frequency of generalized discharges. EM, AA. Self stimulation wirh eyes closure.			Sleep: Multifocal spikes, prominent on frontal and occipital regions; bursts of generalized irregular polyspike or PSW. EM, EMA, AA, AAOC, AS, MS, AAA. Self stimulation wirh eyes closure.						Sleep: Low-voltage centrooccipital spikes. AA, AAOC, MA		Wake: Diffuse GPSW; Sleep: PSW on frontal regions. EM, EMA, AA, AAM, MS, AS, MATS. Self stimulation wirh eyes closure.				Wake: Diffuse SW, predominant on frontal regions; Sleep: Numerous frontal SW. EMA, AAM		Wake: Temporo-parietal SW; Sleep: PSW on frontal regions
MRI	Cranial		Normal			NR			Normal		Normal			Bilateral hypersignal of WM (4 yr); cerebellar atrophy (6 yr)						Normal		Aspecific WM hypersignal				NR		Defect in frontal lobes develoment
Treatment	AED		VPA, LEV, ETX, LTG, CBD			ETX, LEV, LTG, VPA, CLB, ZNS, PER, CBD			VPA, ETX, ZNS, LTG		LEV, VPA			VPA, CLB, ETX, LTG, RUF, ZNS						LEV, VPA, LTG, ETX		LEV, VPA, TPM, CLB, CLZ, ZNS, LTG, PER				VPA, LEV, TPM		LEV, VPA, LTG, ETX, CLB, CLZ, ZNS
	Other		KD			KD			KD					KD, VNS						KD		KD						
Genetic inform.	Genomic change (Hg19)		chr6:33409002; G>G/T			chr6:33409095; C>C/T			chr6:33409095; C>C/T		chr6:33411544delA			chr6:33405604; T>T/C						chr6:33409002; G>G/T		chr6:33414346; G>G/A				chr6:33411127; A>A/G		chr6:33400531-33400532insG
	cDNA/aa change		c.1966G>T, p.Glu656*			c.2059C>T, p.Arg687*			c.2059C>T, p.Arg687*		c.3215_3224del, p.Lys1072Serfs*2			c.922T>C, p.Trp308Arg						c.1966G>T, p.Glu656*		c.3583-6G>A, p.Val1195Alafs*27				c.2798A>G, p.His933Arg		c.456insG, p.Thr153Aspfs*15
	Inheritance		de novo			de novo			de novo		de novo			de novo						de novo		de novo				de novo		de novo
	Family History		NR			Paternal grandmother with unspecified epilepsy			NR		NR			NR						NR		NR				NR		NR

^1^ Syndromic diagnosis of EMA; ^2^ Syndromic diagnosis of MAE; ^a,b,c,d,e^ Different cases with the same genomic change. aa: amino acid; aCGH: Array comparative genomic hybridization; AS: Angelman syndrome; ASD: Autism spectrum disorder; CSF: Cerebrospinal fluid; DD: Developmental delay; DDD: Deciphering Developmental Disorders; del: deletion; EEG: Electroencephalogram; FOS: Fixation of the sensitivity; ID: Intellectual disability; IPS: Intermittent Photic Stimulation; MRI: Magnetic resonance imaging; NGS: Next generation sequencing; NR: not reported; p: percentile; OCD: Obsessive compulsive disorder; PPR: Photoparoxysmal response; PS: Photosensitivity; REM: Rapid eye movement; VUS: variant of unknown significance; WES: whole exome sequencing; WM: White matter; yr: year; -: Feature not found. Seizure types: AA: Atypical absence; AAA: Atypical absences with atonic phenomena, AAM: Atypical absence with myoclonia; AAOC: Atypical absences with oculoclonic movements; AS: Atonic seizures; DA: drop attack; EM: Eyelid myoclonia; F: focal seizures; FIAS: Focal impaired awareness seizures; FS: Febrile seizures; GS: Generalized seizures; GTCS: Generalized tonic clonic seizures; MA: Myoclonic absence; MAt: myoclonic-atonic seizures; MATS: Myoclonic-atonic-tonic seizures; MFD: multifocal discharges; MJ: Myoclonic jerks; MS: Myoclonic seizures; NCSE: Non-convulsive status epilepticus; TCS: Tonic-clonic seizures; TS: Tonic seizure. EGG: BG: Background; DS: diffuse slowing; ED: epileptiform discharge; FD: focal discharges; GD: General discharges; GPSW: Generalized polyspike wave; GSW: Generalized spike wave, PSW: polyspike wave; SSW: slow spike and wave; SW: spike wave. Treatment: CBD: cannabidiol; CBZ: carbamazepine; CLB: clobazam; CLZ: clonazepam; KD: Ketonic diet; ETX: ethosuximide; LCM: lacosamide; LEV: levetiracetam; LTG: lamotrigine; LZP: Lorazepam; AD: modified Atkins diet; NZP: nitrazepam; PER: perampanel; PHT: Phenyltoin; RUF: Rufinamide; TPM: topiramate; VGB: vigabatrin; VNS: vagal nerve stimulation; VPA: valproate; ZNS: zonisamide.

**Table 2 ijms-22-05609-t002:** Summary of the cases reported with a pathogenic (or probably pathogenic) alteration in *NEXMIF* (NM_ 001008537.3) and an EMA/EMA-like phenotype.

	A													
	Reference	Samanta 2020 ^1^		Wu 2020		Stamberger 2021 ^1,2^		Stamberger 2021 ^1,2^	Stamberger 2021 ^1,2^	Stamberger 2021 ^1^		Stamberger 2021 ^1^		Stamberger 2021 ^1,2^/Myers 2018 ^2^
	Patient					1 (F1)		2 (F2)	4 (F4)	7 (F5)		8 (F6)		10 (F7)/T990 (family 12)
Clinical features	gender	Female		Female		Male		Male	Female	Female		Female		Female
	Age	9 yr		29 yr		8 yr		12 yr	12 yr	14 yr		18 yr		28 yr
	DD/ID	+ (mild)		+ (mild)		+ (severe)		+ (moderate)	+ (severe)	+ (moderate)		+ (moderate)		+ (moderate-severe)
	Behavioral features	ADHD		NR		Regression, ASD, behavioral problems		Regression, ASD, severe tantrums	Regression, ASD, ADHD	Regression, obsessive, repetitive behaviours, anxiety		Regression, stereotypes, agressive behaviour, impulsive, attention problems, anxiety		ASD traits
	Other parametres	mild hypotonia		Minor dysmorphic features (flat nasal bridge and ocular hypertelorism), diabetes mellitus type 2		Scaphocephaly, mild facial dysmorphisms (deep set eyes, wide spaced teetch, prominent lower lips, protruding tongue; tapering fingers), hypotonia/hupertonia, esotropia		NR	Upslanting palpebral fissures, hypoplastic eyelashes, small rounded nasal tip	ventricular septal defect; primary enuresis		NR		overweight, prominent eyebrows, hirsuitism, polycystisc ovarial syndrome
Epilepsy	Age of onset	2 yr		6 yr		21 months		15 months	12–14 months	2 yr		3 yr		19 months
	Abscence seizures	+		+ (AAS)		+		+	+	+		+		+
	Eyelid myoclonia	+		+ (rare)		+		+	+	+		+		+
	Photosensitivity	+		+		+		-	-	-		+		-
	Other seizures	Rapid eye blinking with upward eye rolling associated with head bobbing		GTCS, brief blanking, behavioral arrest and states of prolonged confusion		AS, MS. Triggered by PS		MS, MAS, GTCS, NCSE. Triggered by temperature (hot)	Head nods, drop attack, AS, MS, likely NCSE. Triggered by fever, temperature, eye closure	MS, NCSE (absence status)		nocturnal MS, rare GTCS		MS, DA, GTCS, NCSE
EEG	PPR	+		+		NR		NR	NR	NR		NR		NR
	others	3 Hz GSW. Eyelid jerking, generalized epileptiform discharges induced by eye cosure		Interictal: paroxysmal GPSW induced by eye closure, PS. Ictal: persistent 1.5–2.5 Hz semi-rhythmic GPSW. Epileptiform discharges induced by eye closure		Interictal: BG slowing, 2.5–3 Hz GPSW, PFA. Ictal: 2.5 Hz GPSW. Triggered by sleep abd IPS		Interictal: BG slowing, 2.5–4 Hz G(P)SW. Ictal: 2.5–4 Hz GPSW (MS, MAS, head drops). Triggered by sleep, eye closure	Interictal: BG slowing, G(P)SW. Ictal: irregular G(P)SW (eyeclosuse with EM). Triggered by eye closure, posible IPS	Interictal: excessive beta activity, G(P)SW. Ictal: irregular GSW (A-EM, MS). Triggered by eye closure, IPS		Interictal: BG slowing 2.5–5 Hz G(P)SW, MFD. Ictal: G(P)SW (EM, MS), NCSE in sleep. Triggered by sleep, eye closure, IPS		Interictal: BG slowing, G(P)SW, PFA, left rhythmic delta activity, MFD
MRI	Cranial	Normal		Normal		Short medulla, thin, dysmorphic CC, asymmetrical hippocampi, right incompletely rotated, delayed myelination subcortical WM and increased FL/T2 signal in posterior PVWM		Slightly small cerebellar vermis and mild tonsillar ectopia, bulky amygdalas and hippocampal heads.	Normal	Possible minimal atrophy superior cerebellar vermis		Normal		Normal
Treatment	AED	ETX, LTG, MPH, GF, RD, LEV, VPA, RUF, TPM, ZNS		CBZ, VPA, MZN, LTG, LEV, TPM		VPA, LTG, LEV		ETX, VPA, FBM, RUF, TPM, CLB, CBZ, LEV, PLP, CLZ, VGB, LTG, GB, CS	CS, CLB, LEV, VPA.	ETX, ZNS, PB, CS, VPA, LTG, AZA, CLB, LEV, ATD, CBD		CS, NZP, LTG, CBZ, VPA, TPM, CLB, ZNS, ETX		CS, VPA, LEV, LCM, TPM, AZA, PB, ETX, LTG, PER, BRV, CBD
	Other	KD, mAD, LHID, DAP		GPO, STG				KD		KD		KD		
Genetic information	Genetic test	NGS panel (1148 genes)		WES		NR		NR	NR	NR		NR		NR
	Genomic change (Hg19)	chrX:73962671_73962674 del		ChrX:73963328; AG>AG/A-		chrX:73963494; C>C/A		chrX:73961747; G>G/C	chrX:73962951;G>G/A	chrX:73962417; G>G/A		chrX:73961593; G>G/T		chrX:73963428; G>G/A
	cDNA/aa change	c.1718_1721delATCA, p.Asp573Serfs*11		c.1063delC, p.Leu355*		c.898G>T, p.Glu300*		c.2645C>G, p.Ser882*	c.1441C>T, p.Arg481*	c.1975C>T, p.Gln659*		c.2799C>A, p.Tyr933*		c.964C>T, p.Arg322*
	Inheritance	*de novo*		*de novo*		*de novo*		NR	inherited (maternal)	*de novo*		*de novo*		inherited (paternal gonadal mosaicism likely)
	Others information	CXI 74:26 (random). CSF GLUT1 and aCGH normal		CXI 51:49 (random)		NR		~30% mosaicism for *NEXMIF* alteration	CXI ~90:10 (skewed). SCN1A:p.(Met1977Val), paternal - VUS	CXI~60:40 (random)		CXI~50:50 (random)		CXI~80:20 (skewed)
	Family History			No family history of IDnor epilepsy		Family history of GEFS+ and hypotonia: Father: seizures, hypotonia, speech/language delay, unilateral hearing loss. Sister: FS. Paternal grandfather, paternal aunt and two paternal uncles: childhood epilepsy +- FS. Paternal cousin: FS		Maternal great-great-grandmother: epilepsy	Two sisters carriers of the alteration with ID but without seizures (patient 5 and 6). Two more affected siblings not included in the study with ID without seizures. One other sister is carrier with no disease activity to date. Carrier mother has mild ID					Epileptic sister, also with MAE, carrier of the alteration (patient 9).
	**B**													
	**Reference**	**Stamberger 2021 ^1,#^**	**Stamberger 2021 ^1,#^**		**Stamberger 2021 ^1^**		**Stamberger 2021 ^1,&^**	**Stamberger 2021 ^1^**	**Stamberger 2021 ^1^**		**Stamberger 2021 ^1^/Borlot 2017 ^1^**		**Stamberger 2021 ^1^**	**Stamberger 2021 ^1,2,&^**
	**Patient**	**13 (F10)**	**15 (F12)**		**16 (F13)**		**18 (F15)**	**23 (F20)**	**33 (F30)**		**34 (31)/27**		**37 (F34)**	**41 (F38**
Clinical features	gender	Female	Female		Female		Female	Female	Female		Female		Female	Female
	Age	8 yr	12 yr		10 yr		15 yr	16 yr	15 yr		26/23 yr		10 yr	4 yr
	DD/ID	+ (mild)	+ (moderate)		+ (moderate)		+ (mild)	+ (moderate)	+ (moderate)		+ (mild)		+ (moderate)	+ (mild)
	Behavioral features	Aggressive behaviour, attention problems	ADHD		ASD, Agressive behaviour		Self-abasement, ASD traits (social difficulties)	NR	Easily frustrated		Depression, anxiety		Attention deficiency and problems linked to communication difficulties during infancy, decreased satiety, tics (blinking), ASD traits.	-
	Other parametres	overweight, gastro-oesophageal reflux disease	NR		Mild facial dysmorphisms (short philtrum, low-set hairline, mild prognathism with frontal bossing)		NR	Hypotonia, hypermovility	NR		overweight, gastro-oesophageal reflux disease as infant/-		Low set backward rotated ears, protruding underlip, hypotonia	Mild hypotonia and hyperlaxity
Epilepsy	Age of onset	1 yr	9 monts		2–4 months		2–4 yr	18 months	6.5 yr		16 months		30 months	2 yr 10 months
	Abscence seizures	+	+		+		+	+	+		+		+	NR
	Eyelid myoclonia	+	+		+		+	+	+		+		+	+
	Photosensitivity	NR	NR		-		NR	+	-		-		-	+
	Other seizures	MS, GTCS. Triggered by sleep deprivation	Triggered by fever, eye closure		Triggered bu eye closure		GTCS	GTCS, NCSE (absences), Triguered by PS	NR		Single GTCS/BCS		MS	AS (head drops) MS (blinking). Triggered by PS
EEGG	PPR	NR	NR		NR		NR	NR	NR		NR		NR	NR
	others	Interictal: mild BG slowing, >3 Hz G(P)SW, MFD, GPFA. Ictal: G(P)SW (Absences +-EM), GSW (MS). Triggered by sleep, IPS, hiperventilation	Interictal: normal BG, G(P)SW in sleep, MFD. Ictal: GSW (EM), Triggered by sleep, eye closure		Interictal: normal BG, G(P)SW in sleep, MFD. Ictal: 3 Hz irregular GPSW (EM). Triggered by sleep		Interictal: BG asimmetry, near continuous G(P)SW during wakefulness. Ictal: EM with impaired awareness	Interictal: Normal BG, MFD with (P)SW, multiple spikes. Triggered by hyperventilation, IPS, sleep, eye closure	Interictal: G(P)SW, PFA. IctalG(P)SW (EM, MS). Triggered by hyperventilation, IPS, eye closure, fixation of sensitivity		Interictal and ictal: sharply contoured runs of alpha activity at times/polyspike and generalized spike waves induced by eye closure		Interictal: multiple spikes and spike-wave. Ictal: quick frontal and central activity (MS). Triggered by eye closure	Interictal: BG slowing, G(P)SW, MFD, bifrontal disrythmic delta activity during sleep. Ictal: GPSW (MS). Triggered by sleep
	Cranial MRI	Normal	Normal		Normal		Normal	Normal	Normal		Normal		Normal	Normal
Treatment	AED	ETX, LEV, VPA	LTG, CLZ, LCM, VPA, ETX, CLB, LEV		ETX, CLZ, VPA		VPA, LTG	LEV, CLZ, ZNS, VPA, ETX, OXC, LTG	VPA, CLB		TPM, CBZ, VPA		LEV, LTG, ETX, VPA	CBD, VPA, CLB, LEV
	Other							KD, VNS			vitamin B6			
Genetic information	Genetic test	NR	NR		NR		NR	NR	NR		aCGH		NR	NR
	Genomic change (Hg19)	chrX:73962510; G>G/A	chrX:73962510; G>G/A		chrX:73961016; C>C/A		chrX:73961500; G>G/C	chrX:73964056; C>C/T	chrX:73963740; G>G/A		ChrX:73930523_74007913 del (0.08 Mb, 1 gene)		chrX:73960934dupT	chrX:73961500; G>G/C
	cDNA/aa change	c.1882C>T, p.Arg628*	c.1882C>T, p.Arg628*		c.3376G>T, p.Glu1126*		c.2892C>G, p.Tyr964*	c.336G>A, p.Trp112*	c.652C>T, p.Arg218*		Xq13.3 del (Ex 2–4)		c.3458dupA, p.Asn1153Lysfs*8	c.2892C>G, p.Tyr964*
	Inheritance	NR	*de novo*		*de novo*		*de novo*	*de novo*	*de novo*		*de novo*		*de novo*	NR
	Others information		CXI~65:35 (random)		CXI~65:35 (random)			SMA: 3p24.1(30,414,405–30,878,291)x3, maternal - VUS ADGRV1: p.(Asp2942His), maternal - VUS	CXI ~60:40 (random)		GRIN2A: p.(Asn106Lys), paternal - VUS/GLUT1 deficiency (SLC2A1 sequencing) and Epilepsy panel (476 genes) normal		CXI ~50:50 (random)	
	Family History										Maternal distant cousin: GTCS and learning disability. Distant maternal relative: absence seizures in childhood. Sister of paternal grandmother: "drop seizures" and questionable DD		Sister with normal development exhibited epilepticus status due to a fall in bicycle, attention deficiency	

^1^ Syndromic diagnosis of EMA; ^2^ Syndromic diagnosis of MAE; ^#,&^ Different cases with the same genomic change. aa: amino acid; aCGH: Array comparative genomic hybridization; ADHD: Attention deficit hyperactivity disorder; AED: Antiepileptic drug; ASD: Autism spectrum disorder; CC: corpus callosum; CSF: Cerebrospinal fluid; CXI: X-inactivation; DD: Developmental delay; del: deletion; EEG: Electroencephalogram; Ex: exon; ID: Intellectual disability; IPS: Intermittent Photic Stimulation; MRI: Magnetic resonance imaging; NGS: Next generation sequencing; NR: not reported; PPR: Photoparoxysmal response; PS: Photosensitivity; PVWM: Periventricular white matter; VUS: variant of unknown significance; WM: White matter; yr: year; -: Feature not found. Seizure types: AAS: Atypical absence status; AS: Atonic seizures; DA: drop attack; EM: Eyelid myoclonia; FS: Febrile seizures; GTCS: Generalized tonic clonic seizures; MAS: Myoclonic-atonic seizures; MFD: multifocal discharges; MS: Myoclonic seizures; NCSE: Non-convulsive status epilepticus; PFA: paroxysmal fast activity. EGG: BG: Background; GPFA: Generalized paroxysmal fast activity; GPSW: Generalized polyspike wave; GSW: Generalized spike wave, PFA: paroxysmal fast activity. Treatment: AZA: acetazolamide; ATD: Amantadine; BRV: brivaracetam CBD: cannabidiol; CBZ: carbamazepine; CLB: clobazam; CLZ: clonazepam; CS: corticosteroids; KD: Ketonic diet; DAP: dexamphetamine; ETX: ethosuximide; FBM: felbamate; GB: gabapentin; GF: guanfacine; GPO: glimepiride; LCM: lacosamide; LEV: levetiracetam; LHID: low hypoglycemic index diet; LTG: lamotrigine; mAD: modified Atkins diet; MPH: methylphenidate; MZN: midazolam; NZP: nitrazepam; OXC: oxcarbazepine; PB: phenobarbital; PER: perampanel; PLP: Piridoxal phosphate; RD: risperidone; RUF: Rufinamide; STG: sitagliptin; TPM: topiramate; VGB: vigabatrin; VNS: vagal nerve stimulation; VPA: valproate; ZNS: zonisamide.

**Table 3 ijms-22-05609-t003:** Summary of the cases reported with a pathogenic (or probably pathogenic) alteration in *RORB* (NM_006914) and an EMA/EMA-like phenotype.

	Reference	Bartnik 2012	Rudolf 2016 ^1^	Rudolf 2016 ^1^	Rudolf 2016 ^1^	Rudolf 2016 ^1^	Rudolf 2016	Rudolf 2016	Sadleir 2020 ^1^	Sadleir 2020	Morea 2021 ^1^
	Patient	12	4	13	14	20	9A1117	GE0705	Familly C II-2	Family D II-1	Case report
Clinical features	gender	Male	Female	Female	Female	Female	Female	Female	Female	Male	Male
	Age	NR	NR	NR	NR	NR	25 yr	10 yr	40 yr	20 yr	21 yr
	DD/ID	NR	+ (mild)	+ (mild)	+ (mild)	NR	+ (mild)	+	NR	-	+ (moderate)
	Behavioral features	ASD	NR	NR	NR	NR	NR	NR	NR	NR	ADHD
	Other parametres	NR	NR	NR	NR	NR	Macrocephal, overweight, and learning dificulties	convergent strabismus, hypermetropia, learning dificulties	Learning dificulties	Learning dificulties	NR
Epilepsy	Age of onset	2 yr	13 yr	3 yr	9 yr	11 yr	5 yr 5 months	4 yr 9 month	3 yr	10 yr	4 yr
	Abscence seizures	NR	+	+	+	+	+ (TA, absence status)	+	+	+ (absence status)	+
	Eyelid myoclonia	+	+	+	+	+	+	+	+	+	+
	Photosensitivity	NR	+	+	+	+	-	+	+	+	NR
	Other seizures	GTCS	GTCS	GTCS	GTCS	GTCS	GTCS, FS, TC	nocturnal GTCS	GTCS, occipital seizures	GTCS, occipital seizurures,	Induced by television and videogame exposure
EEG	PPR	NR	+	+	+	+	NR	NR	+	+	NR
	others	CTS	3 Hz GSW	3 Hz GSW	3 HZ GSW	3 Hz GSW	Interictal: normal BG rhythm and bilateral centrotemporal spikes. 3 Hz SW absence seizures activated by hyperventilation. Ictal Absence seizures. Interictal GSW, GPSW. Focal frontal or temporal occipital paroxysmal activity	Absence seizures ocassionally with EM triggered by IPS. 3 Hz GSW	GSW	GSW	Ictal: 3 Hz GSW. Interictal: asynchronous spikes on a physiological BG rhythm. Generalized epileptic discharges elicited by eye closure and hyperventilation
	Cranial MRI	NR	NR	NR	NR	NR	Normal	Normal	-	Normal	Normal
Treatment	AED	NR	3 treated with VPA, one with ETX and PB				CBZ, VPA, ETX, VGB, CLB, LTG, TPM, LEV	TEX, VPA, LTG, LEV	LTG, VPA	VPA, LEV, LTG	VPA, ETX, LEV
	Other							KD			
Genetic information	Genetic test	aCGH	WES	WES	sanger sequencing	WES	aCGH	aCGH	aCGH	Sanger sequencing	NGS
	Genomic change (Hg19)	chr9:7474,400_ 77306932 del (2.57 Mb, 6 genes)	chr9:77249649; C>C/T				chr9:70984481_ 79549501 del (8.5 Mb, 47 genes)	chr9:77261322_ 77313598 del (52Kb)	Min:chr9:77249078_ 77251973del2896 Max:chr9.77248677_77251984del3308	chr9:77286755; A>A/T	9q21
	cDNA/aa change		c.196C>T, p.Arg66*					9q31.13 del (Ex 5–10)	c.96_237del141, p.Gly32_Ala79del48	c.1162A>T, p.Ile388Phe	NR
	Inheritance	de novo	(probably) Inherited				de novo	de novo	inherited (maternal)	inherited (paternal)	de novo
	Others information	FISH: 9q21.13 (RP 11-243A1)		aCGH array normal			Karyotype: mos 47,XX,+r [20]/46,XX,-21,+der(9)t(9;21) [5]/46,XX [25]. aGCH array: mosaic gain 9p. FISH: 9q13q21.13 del (RP11-404E6), mosaic i(9p), der(9)t(9;21) and r(9)	qPCR to validate the result	NR	NR	
	Family History	Father Asperger syndrome	Family members: Patient 20; Patient 13 (Mother); Patient 14 (maternal aunt); Patient 4 (maternal grandmother). Other two carriers: Patient 23 (sister): One episode of absence seizure (probably GGE with no EEG); Patient 10 (maternal great-aunt): EEG with isolated high-amplitude spike during IPS but seizure state not confirmed. Several antecedents of PS without seizures				Maternal uncle and two of her first cousins had GTCS	No family history of seizures	Alteration inherited from her mother (normal intelligence and no seizures). Her son, with intratable DEE and severe ID, has the same microdeletion	Alteration inherited from his father, diagnosed with early onset abcense epilepsy and occipital lobe epilepsy, who also presented GTCS	-

^1^ Syndromic diagnosis of EMA. aa: amino acid; aCGH: Array comparative genomic hybridization; ADHD: Attention deficit hyperactivity disorder; AED: Antiepileptic drug; ASD: Autism spectrum disorder; DD: developmental delay; DEE: developmental and epileptic encephalopathy; del: deletion; EEG: Electroencephalogram; Ex: exon; GGE: genetic generalized epilepsy; ID: Intellectual disability; IPS: Intermittent Photic Stimulation; MRI: Magnetic resonance imaging; NGS: Next generation sequencing; NR: not reported; PPR: Photoparoxysmal response; yr: year; -: Feature not found. Seizure types: FS: Febrile seizures; GTCS: Generalized tonic clonic seizures; TA: typical absence; TC: tonic-clinic seizure. EGG: BG: background; CTS: centro-temporal spike; GPSW: Generalized polyspike wave; GSW: Generalized spike wave. Treatment: CBZ: carbamazepine; CLB: clobazam; ETX: ethosuximide; KD: Ketonic diet; LEV: levetiracetam; LTG: lamotrigine; PB: phenobarbital, TPM: topiramate; VGB: vigabatrin; VPA: valproate.

**Table 4 ijms-22-05609-t004:** Summary of the cases reported with a pathogenic (or probably pathogenic) alteration in *CHD2* (NM_001042572) and an EMA/EMA-like phenotype.

	Reference	Galizia 2015 ^1^	Galizia 2015 ^1^	Galizia 2015 ^1^	Tomas 2015 ^1^/Carvill 2013 ^2^	Tomas 2015 ^1^	Tomas 2015 ^1^	Tomas 2015 ^1^/Mullen 2013
	Patient	7	8	9	5/T38	6	8	9/15 [57]
Clinical features	gender	NR	NR	NR	Male	Female	Male	Female
	Age	NR	NR	NR	18/17 yr	13 yr	14 yr	36/26 yr
	DD/ID	NR	NR	NR	+ (moderate-severe/moderate)	+ (moderate-severe)	+ (mild)	+ (mild)
	Behavioral features	ASD	NR	NR	ASD, regression, aggression/ASD, No regression	ASD, ADHD, aggresion	ADHD, regression, agression	autistic traits, regression, agression
	Other parametres	nephrolithiasis, migraine, scoliosis	NR	NR	Transient ataxia on valproate	Short stature	Short stature, ataxia	NR
Epilepsy	Age of onset	NR	NR	NR	12 months	30 months	30 months	34 months
	Abscence seizures	+	+	+	+ (AMA, MA; TA)/+	+ (TA)	+	+
	Eyelid myoclonia	+	+	+	+	+	+	+
	Photosensitivity	+	+	+	+ (Self induced with TV)	+ (Self induced with TV)	+(Self induced with TV; photic stimulation)	+(Self induced with TV or light)
	Other seizures	NR	NR	NR	MS, FS, GTCS/AS, FS, MJ, TC	MS (self-induced with TV), TS, GTCS	AS, GTCS, MS (self-induced with TV), NCSE, CSE	GTCS, MS (self-induced with TV or light),
EEG	PPR	+	+	+	- (interictal and BG)	- (interictal and BG)	+ (grade 4)	NR
	others	NR	NR	NR	3–4-HZ GPSW during atonic myoclonic absence seizures. 9 yr: Diffusely slow BG with symmetrical theta and 3 Hz delta. 14 yr: Normal BG for brief bursts of 3–4 Hz regular rhythmic GSW/3.8 Hz GSW	3 yr: Slow BG for age. Inter-ictal GSW GPSW, bi frontal slow spike wave (2 Hz). Eye flickering—bursts of 2 Hz bifrontal spike and wave. 4 yr: EMA, GSW, GPSW. Activated by eye closure. 7 yr: Normal BG, irregular GSW with a frontal predominance. GSW on eye closure and during eyelid myoclonia	32 months: Normal BG, frontal predominant GSW, GPSW. 5.5 yr: Normal BG, multifocal GSW. 11 yr: Diffusely slow BG, 3 Hz GSW, bifrontal spikes, activated in sleep, 4–5 Hz posteriorly dominant GSW on eye closure. 12 yr: Continuous slowing in wake and marked generalized epileptiform activity in sleep. 14 yr: Diffuse theta slowing, active interictal GSW GPSW discharges >3 Hz increased by hyperventilation	5 yr: Generalized epileptiform discharges every 1–2 minutes
MRI	Craneal	NR	NR	NR	Normal	Generalized cerebellar atrophy with large v4 and prominent folia. The posterior corpus callosum is foreshortened and smaller posteriorly	Atrophy between scans, markedly in the cerebellum. The corpus callosum is hypoplastic posteriorly with a small splenium	Normal
Genetic data	Genetic test	NGS	NGS	NGS	target NGS	target NGS	target NGS	aCGH
	Genomic change (Hg19)	chr15:93540316; A> A/-	chr15:93545442; ->-/A	chr15:93482909; C>C/T	chr15:93545504_93545507del	chr15:93557956delG	chr15:93521611; C>C/T	chr15:91027533_93477874 del
	cDNA/aa change	c.3725delA, p.Lys1245Asnfs*4	c.4173dupA, p.Gln1392Thrf*17	c.C653T, p.Pro218Leu	c.4235_4238 delAAGG, p.Glu1412Glyfs*64	c.4720delG, p.Gly1575Valfs*17	c.2725C>T, p.Gln909*	15q26 del (2.4 Mb)
	Inheritance	NR	de novo	inherited	de novo	de novo	de novo	de novo
	Family History	NR	NR	Inherited from unaffected mother	NR	NR	NR	NR

^1^ Syndromic diagnosis of EMA; ^2^ Syndromic diagnosis of MAE. aa: amino acid; aCGH: Array comparative genomic hybridization; ADHD: Attention deficit hyperactivity disorder; AED: Antiepileptic drug; ASD: Autism spectrum disorder; DD: developmental delay; del: deletion; EEG: Electroencephalogram; ID: Intellectual disability; MRI: Magnetic resonance imaging; NGS: Next generation sequencing; NR: not reported; PPR: Photoparoxysmal response; yr: year. Seizure types: AMA: Atonic myoclonic absence; AS: Atonic seizure; CSE: Convulsive status epilepticus; FS: Febrile seizures; GTCS: Generalized tonic clonic seizures; MA: Myoclonic absence; MJ: Myoclonic jerks; MS: Myoclonic seizures; NCSE: Non-convulsive status epilepticus; TA: typical absence; TC: tonic-clinic seizure, TS: Tonic seizure. EGG: BG: Background; GPSW: Generalized polyspike wave; GSW: Generalized spike wave.

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
