# Peer review of "Candidate Genes for Eyelid Myoclonia with Absences, Review of the Literature"

_ijms, 2021, doi:10.3390/ijms22115609_

Round 1

Reviewer 1 Report

In this review article, authors summarized studies with seven human genes associated with Eyelid Myoclonia with Absence (EMA) and EMA-like phenotype. This is a nice compilation of studies related to EMA.

I can make a few suggestions to improve the manuscript.

  • Before publication, the text should carefully be edited. Some examples are described below.

Line 41: “EEG” should be “Electroencephalogram (EEG)”

Line 54: “there is no know gene” should be “there is no known gene”.

Line 55: “only different case reports have propose several candidate genes” should be “only different case reports have proposed several candidate genes”.

Line 97, 208, 269, 303: “EGG: Electroencephalogram” should be “EEG: Electroencephalogram”.

Although authors focused on the gene diagnosis studies with human patients in this manuscript, I would like to encourage them adding an additional section summarizing rodent models targeting some of these seven genes, such as SYNGAP1 and NEXMIF. That would also be informative to the readers.

Reviewer 2 Report

Major point:

1) SLC2A1, KCNB1 and NAA10 could only very optimistically be considered as candidate genes. Their candidacy needs more robust experimental data. As only very few studies (one, perhaps two) exist for each of these genes, the authors should consider their omission from this otherwise excellent survey of functional and genetic data. In harmony with the authors’ note in lines 456/457 “There is no enough information to establish a clear relationship...”, discussing the four “strongest” cases would perhaps increase the impact of the manuscript.

There are several minor points the authors should address:

2) The number of patients used to analyze data for each gene should be clearly stated in the Methods section. Some of these data are dispersed in the Results section.

3) It should be clearly indicated how many cases were included for each gene or when data from the same patient were used in different studies (e.g., patient 26/ T2528 in the studies reported by Vlaskamp et al., 2019 and Carvill et al., 2013, respectively; see Table 1).

4) Ref. 21. and Ref 28. have Von Sülpnagel’s name differently spelled.

5) “Array comparative genomic hybridation” should be changed to “Array comparative genomic hybridization” throughout the manuscript.

6) line 106: “lorazepamm” should be changed to “lorazepam”

7) lines 110 and 145/146 seem to contradict each other; please clarify.

8) line 396: “NEFMIF” should be changed to “NEXMIF”

9) line 456: “... no enough...” should be changed “... not enough...”
